# Chronic Pulmonary Aspergillosis: Notes for a Clinician in a Resource-Limited Setting Where There Is No Mycologist

**DOI:** 10.3390/jof6020075

**Published:** 2020-06-02

**Authors:** Felix Bongomin, Lucy Grace Asio, Joseph Baruch Baluku, Richard Kwizera, David W. Denning

**Affiliations:** 1Department of Medical Microbiology & Immunology, Faculty of Medicine, Gulu University, Gulu P.O. Box 166, Uganda; asiolucia@yahoo.com; 2Division of Pulmonology, Mulago National Referral Hospital, Kampala P.O. Box 7051, Uganda; bbjoe18@gmail.com; 3Translational Research Laboratory, Infectious Diseases Institute, College of Health Sciences, Makerere University, Kampala P.O. Box 22418, Uganda; kwizerarichard@ymail.com; 4The National Aspergillosis Centre, Wythenshawe Hospital, Manchester University NHS Foundation Trust, Manchester M23 9LT, UK; david.denning@manchester.ac.uk; 5Division of Infection, Immunity and Respiratory Medicine, School of Biological Sciences, Faculty of Biology, Medicine and Health, The University of Manchester, Manchester M13 9PL, UK

**Keywords:** chronic pulmonary aspergillosis, mycoses, lung disease, resource-limited setting, epidemiology, chest X-ray, *Aspergillus* IgG

## Abstract

Chronic pulmonary aspergillosis (CPA) is a spectrum of several progressive disease manifestations caused by *Aspergillus* species in patients with underlying structural lung diseases. Duration of symptoms longer than three months distinguishes CPA from acute and subacute invasive pulmonary aspergillosis. CPA affects over 3 million individuals worldwide. Its diagnostic approach requires a thorough Clinical, Radiological, Immunological and Mycological (CRIM) assessment. The diagnosis of CPA requires (1) demonstration of one or more cavities with or without a fungal ball present or nodules on chest imaging, (2) direct evidence of *Aspergillus* infection or an immunological response to *Aspergillus* species and (3) exclusion of alternative diagnoses, although CPA and mycobacterial disease can be synchronous. *Aspergillus* antibody is elevated in over 90% of patients and is the cornerstone for CPA diagnosis. Long-term oral antifungal therapy improves quality of life, arrests haemoptysis and prevents disease progression. Itraconazole and voriconazole are alternative first-line agents; voriconazole is preferred for patients with contra-indications to itraconazole and in those with severe disease (including large aspergilloma). In patients co-infected with tuberculosis (TB), it is not possible to treat TB with rifampicin and concurrently administer azoles, because of profound drug interactions. In those with pan-azole resistance or intolerance or progressive disease while on oral triazoles, short-term courses of intravenous liposomal amphotericin B or micafungin is used. Surgery benefits patients with well-circumscribed simple aspergillomas and should be offered earlier in low-resource settings.

## 1. Introduction

*Aspergillus* is one of the oldest known genera of fungi first described by a Roman Catholic clergyman and biologist Pier Antonio Micheli in 1729 [1]. To date, over 330 species of *Aspergillus* have been described [2]. Of these, approximately 50 species are recognised to be pathogenic to humans, and the five most clinically relevant include *Aspergillus fumigatus*, *A. flavus*, *A. niger*, *A. terreus*, and *A. nidulans*, in order of frequency [3,4]. Aspergillosis is a generic term used to describe almost all diseases caused by the opportunistic fungi of the genus *Aspergillus* [5]. Aspergillosis is acquired by way of inhalation or traumatic inoculation of *Aspergillus* spores from dead and decaying organic matter in the environment [6,7]. Human-to-human or animal-to-human transmissions have not been reported [5].

Aspergillosis syndromes are immensely diverse. These syndromes can be also quite fluid if the individual’s immune status changes. While the immune status dictates what form of aspergillosis the patient is at risk of, it also dictates whether or not they are likely to get aspergillosis at all. The following syndromes have been described: (1) acute invasive sinopulmonary disease occurring in severely immuncompromised patients such as those with haematological malignancies, post-transplantation immunosuppression, graft-versus-host disease (GvHD), chronic granulomatous disease, intensive care, decompensated liver cirrhosis and HIV/AIDS; (2) sub-acute invasive and chronic aspergillosis, observed in patients with poorly controlled diabetes, alcohol excess and prolonged systemic corticosteroid therapy (for sub-acute invasive), or those with underlying structural diseases of the lungs and sinuses with impaired physiological or anatomic barriers of innate immunity (for chronic); (3) allergic rhinosinusitis and hypersensitivity lung diseases, observed in patients with atopy, asthma or cystic fibrosis, and (4) mucocutaneous and subcutaneous forms of aspergillosis such as onychomycosis, keratitis and Madura foot which follow traumatic inoculation of the spores directly into the affected tissues [8,9]. Since *Aspergillus* species are ubiquitous in the environment and humans inhale *Aspergillus* spores on a daily basis, sinopulmonary diseases are the most common manifestations of aspergillosis [10,11].

Three major *Aspergillus*-related bronchopulmonary syndromes account for the major causes of morbidity and mortality due to aspergillosis globally [12,13,14]: (1) allergic bronchopulmonary aspergillosis (ABPA), which is an eosinophilic hypersensitivity lung disease classically seen in patients with worsening symptoms of asthma, chronic obstructive pulmonary disease (COPD) and cystic fibrosis; (2) invasive pulmonary aspergillosis (IPA), which is an acute life-threatening disease mainly seen in patients with severe immunosuppression such as those on corticosteroids, and (3) chronic pulmonary aspergillosis (CPA), which is a progressive debilitating parenchymal lung disease manifesting in patients with pre-existent cavitary, structural lung disease or those with subtle immune defects. In addition, a small group of patients, especially those with moderate immunosuppression (poorly controlled diabetes, alcohol excess and liver cirrhosis) develop sub-acute invasive aspergillosis (SAIA)—a less active invasive disease that progresses over days to weeks (<3 months) with a propensity of transforming to IPA or CPA (Figure 1).

This narrative review highlights some of the essential information required by a non-specialist mycologist/clinician in a resource-limited setting: this clinician will occasionally encounter patients with CPA in their practice and will need this information to recognise patients at risk and diagnose and appropriately manage them.

## 2. Definition and Pathogenesis

CPA is a progressive respiratory syndrome developing over several months to years, predominantly in patients with prior or current lung lesions—mainly cavities [15]. CPA is radiologically characterized by one or more cavities with or without a fungal ball present or nodules progressing to lung parenchymal or pleural fibrosis [15,16,17].

The pathogenesis of CPA is fairly well understood, especially for patients with cavitary lung diseases. Inevitable repeated exposure of persons to *A. fumigatus* conidia, the most frequent etiologic agent of CPA, and the small diameter (3–5 μm) of these conidia facilitate their penetration into the alveoli spaces, leading to saprophytic colonization of lung cavities [18]. This may lead to local inflammation, pleural and/or parenchymal fibrosis, expansion of the colonised cavity or creation of new cavities with or without an aspergilloma (also known as a fungal ball—a complex conglomeration of fungal mycelia, fibrin, mucus, inflammatory cells and tissue debris) [5]. Fungal balls form from growth inside a pulmonary cavity that detaches from the cavity wall [19]. The role of genetic aberrations and immune dysregulation in the pathogenesis of CPA and its progression remains a subject of further investigation [20,21].

## 3. Epidemiology

CPA is an emerging fungal infectious disease of public health importance [22]. Globally, it is estimated that over 3 million people suffer from CPA [12]. Of these, approximately 1.2 million cases are thought to be due to previously treated pulmonary tuberculosis (PTB) [23] and over 70,000 cases in patients with sarcoidosis [24]. Thus, PTB and emphysema are the most important risk factors for CPA, and the burden of CPA appears to be higher in areas of high burden of PTB (Table 1). Besides, CPA and PTB can co-exist, posing a challenge in distinguishing the two clinically. This group of patients is often misdiagnosed and managed as smear-negative PTB based on clinical symptoms and suggestive radiology but without microbiological evidence [25]. CPA is most commonly found amongst the middle-aged and elderly people, males and those with a low body mass index, implicating these characteristics as added risk factors for getting the disease [26,27,28,29].

## 4. Underlying Conditions in Chronic Pulmonary Aspergillosis

The vast majority of patients with CPA have underlying structural lung diseases. In studies, it has been found that prior infection with PTB and non-tuberculous mycobacteria (NTM) are the most common major underlying conditions and risk factors for getting CPA, as well as other conditions like ABPA, COPD and/or emphysema, fibrocavity sarcoidosis, bullae or lung cysts, asthma, lung abscess, pneumothorax, *Pneumocystis* pneumonia, treated lung cancer, etc., all posing a risk in acquiring CPA [15,23,26,27,28,29,31,32,33,34]. These conditions create cavities or bullae that put patients at risk of developing CPA, and unfortunately, many patients are found to have more than one of the conditions or a history of more than one [27].

In rare cases, conditions like rheumatoid arthritis, ankylosing spondylitis, silicosis, pneumoconiosis, incompletely resolved invasive pulmonary aspergillosis (IPA) and hyperimmunoglobulin E syndrome can also cause CPA [35]. Immunocompromised patients like those with diabetes mellitus, alcoholism, individuals receiving glucocorticoids like asthmatics are at risk of developing SAIA, a form of CPA that tends to occur in the immunocompromised [15].

There is evidence to show that genetic defects in innate immune functions like toll-like receptor (TLR) 4, interleukin15, TLR3, TLR10, etc., as well low numbers of CD19, CD56 and/or CD4 cells, are common in patients with CPA: they are therefore risk/predisposing factors for CPA [21,36,37,38]. NTM and COPD are associated with higher mortality rates in CPA [39]. Up to 10% of CPA patients have no identifiable risk factors (Table 2). Perhaps these are the patients who may be worked up for immunological defects.

## 5. Diagnosis: The Clinical, Radiologic, Immunologic and Microbiologic (CRIM) Approach

CPA is an overlooked respiratory disease in which delay in making a correct diagnosis and commencement of an appropriate treatment is associated with increased morbidity and mortality [39]. Early identification of predisposing factors and of the patients at risk will improve treatment outcomes of CPA. Diagnosis of CPA is challenging and requires a systematic approach to assessment and interpretation of findings, both of which are necessary for correct disease classification and selection of targeted antifungal treatment and duration of said treatment [41]. Diagnosis of CPA requires a corroboration of a combination of diagnostic armamentaria i.e., Clinical, Radiological, Immunological and Mycological modalities (CRIM) (Figure 2). Current criteria for the diagnosis of CPA are summarised in Table 3 below. The Global Fungal Infection Forum (GFIF) II criteria established by the Global Action Fund for Fungal Infection (GAFFI) in collaboration with international expert panel was particularly developed for the diagnosis of CPA in resource-constrained settings [32].

There is no single modality diagnostic for CPA. For example, a positive sputum *Aspergillus* culture may mean contamination, colonisation or infection; cavitary lung disease on imaging has a myriad of differential diagnoses, and many infectious and non-infectious diseases can present with constitutional symptoms, cough, sputum production and haemoptysis [44,45]. Therefore, it is important to differentiate CPA from other diseases that have a similar presentation and yet different management and prognosis.

### 5.1. Clinical Diagnosis

There is no known pathognomonic clinical feature of CPA. Individuals with CPA vary in age from young adults to those over 80 years of age, with an unclear predilection to middle-aged and elderly male persons [15,17]. It is noteworthy that the early stages of CPA are typically asymptomatic or clinically subtle, with symptoms appearing in a later phase of the disease [46]. Some patients present after 2-10 years of symptoms [15], typically with overt complications.

CPA characteristically presents with a history of cough (usually productive), haemoptysis and weight loss slowly developing over months (>3 months) to years in an apparently immunocompetent or subtly immunocompromised patient with prior or current lung disease [46]. The haemoptysis can be minor or significant and life-threatening (i.e., greater than 150 mL/day) [47]. Patients may also present with non-specific symptoms such as fatigue, chest discomfort, shortness of breath and, though unusual, fever and night sweats; these systemic symptoms provide a distinction between chronic cavitary pulmonary aspergillosis (CCPA) and a simple aspergilloma in which they do not occur [15,48,49].

### 5.2. Radiology

Computed tomography (CT) scans are beneficial in the characterisation of the precise disease pattern and the extent of parenchymal and pleural involvement. On the other hand, chest X-ray (CXR) is important in ruling out CPA if negative: in one study, absence of cavitation and pleural thickening on chest radiography had 100% negative predictive value for CPA [30]. Commonly observed with these modalities is the highly distinctive aspergilloma: appearing as an upper lobe lesion with an air crescent/cavity surrounded by pleural thickening [46,50]. This radiographic appearance of the simple aspergilloma remains stable over several months [19].

In addition to a fungal ball (aspergilloma) [Figure 3], other radiological features include a cavitary lesion with paracavitary thickening/fibrosis, and/or a new cavity or expanding cavities on serial imaging [26,42]. The most common form of CPA presents as CCPA, which when untreated may progress to chronic fibrosing pulmonary aspergillosis (CFPA). Less common is the *Aspergillus* nodule and single aspergilloma [42]. CCPA initially presents as poorly defined regions of consolidation that on serial radiography and can progress to form clearly defined cavities: these cavities, which may contain fungus balls, debris or fluid, are usually thin-walled with little to no associated pleural thickening, although thick-walled cavities and pleural thickening can also occur [42]. Expansion of existing cavities or formation of new ones over months is characteristic of CCPA in the absence of treatment, and left untreated, CCPA may progress to CFPA [42,46]. CFPA has similar radiographic findings as CCPA, combined with substantial fibrosis; whereas SAIA presents as cavitation and consolidation progressively worsening over weeks to a few months [42].

CPA can also present radiologically as a single or multiple nodules, which may or may not cavitate: they are typically not spiculated [46,51]. Unfortunately, because the radiological finding of nodules is often associated with negative *Aspergillus*-specific IgG/precipitins and an inability to attain a diagnostic biopsy, this presentation and therefore the diagnosis is often missed [16,46]. Radiological appearance of nodules may denote lung carcinoma as a possible differential diagnosis, which needs to be ruled out by histological assessment of a nodule biopsy [51].

Use of nuclear medicine modalities for the diagnosis and monitoring of CPA disease activity is not routine.

### 5.3. Immunological/Serological Tests

Serological diagnosis of CPA relies heavily on the demonstration of *Aspergillus*-specific immunoglobulin (Ig) G antibody or *Aspergillus* precipitins. Raised levels of *Aspergillus*-specific IgG antibodies are almost always found in CPA. *Aspergillus*-specific IgG is positive in over 90% of patients with CPA, while *Aspergillus* precipitins, though done, are less sensitive [52,53]. In this context, different methods are currently being exploited in the detection of *A. fumigatus* antibodies in blood. These tests mainly use principles of enzyme-linked immunosorbent assay (ELISA), immunodiffusion and immunoelectrophoresis.

Several tests that check for *A. fumigatus* IgG and IgM antibodies are commercially available in spite of their incomplete validation. *Aspergillus*-specific IgG levels have been successfully used to monitor response of CPA to medical therapy. At a cut-off value of 27 mgA/L, *A. fumigatus*-specific IgG (Phadia) is a reliable test with high sensitivity and specificity in the diagnosis of CPA [54]. However, similar studies have shown that using the ImmunoCAP system, a cut-off of 40 mg/L appears sub-optimal for CPA diagnosis and may require revision in this context [55]. Commonly used tests are the enzyme-linked immunosorbent assay (ELISA) and a new Lateral Flow Device (LFD) (Figure 4).

ELISA has been used for detection of *Aspergillus*-specific IgG antibodies in patients for years, with the most common types used being the competitive and the sandwich type. For diagnosis of CPA, ELISA can be completely automated or done manually but most are done manually [52,55,56,57]. ELISA specificity and sensitivity in testing varies depending on the manufacturer, with the preferred recommended assay sensitivity and specificity being 90% and 85%, respectively, for more accurate results [32]. LFD tests have been around for years and have been used in different medical settings, for instance, in the diagnosis of rheumatoid arthritis and as pregnancy test kits [58,59]. They are now available for *Aspergillus*-specific antibodies detection, with the LDBio *Aspergillus* immunochromatographic LFD being the only commercially available one; this specific assay possesses an 88.9–91.6% sensitivity and 96.3–98% specificity, comparable with ELISA tests presently in use [60,61]. LFD tests are advantageous in that they are inexpensive, easy to use, point-of-care diagnostic methods that have a very short turnaround time compared to other related diagnostic methods; they are therefore ideal for use in resource-limited settings and ease diagnostic testing. [46,62]. LFD tests therefore meet the World Health Organization ASSURED criteria for being Affordable, Sensitive, Specific, User-friendly, Rapid and robust, Equipment-free, and Deliverable to end users, even though they have not yet been used quantitatively or semi-quantitatively. [63] *Aspergillus*-specific IgG LFD should therefore be considered and listed on the WHO essential diagnostic list, as it has a particular unmet need in low- and middle-income countries (LMICs).

Notably, most *Aspergillus* IgG tests detect *A. fumigatus* IgG because *A. fumigatus* is the most common cause of CPA [33]. Therefore, in cases where a patient with highly suspicious findings has a negative *Aspergillus* IgG test, it is recommended that an alternative *Aspergillus* specific IgG for non-*fumigatus Aspergillus* species be done, as other species like *A. flavus*, *A. terreus* and *A. niger* can also cause CPA [35,64].

### 5.4. Microbiological

Clinically or radiologically diagnosed CPA requires microbiological investigations to confirm the diagnosis. These include serology as described above, direct microscopy and fungal cultures and identification, as well as antifungal susceptibility testing [65]. Direct microscopy allows rapid presumptive identification of fungal pathogens usually to genus level, and it is from direct microscopy that the decision of particular culture media and subsequent procedures is made. Presumptive identification also guides selection of appropriate antifungal agents. However, culture of respiratory samples should be performed irrespective of the results of microscopy findings. The method involves mounting the clinical specimen on a microscopic glass slide with 10% potassium hydroxide (KOH) [31]. Conventional culture of *Aspergillus* species from sputum or bronchoalveolar lavage fluid (BALF) samples has low to medium sensitivity of about 10–40%, with a turnaround time of about 3–14 days. Galactomanann is a cell wall constituent of Aspergillus and other fungal pathogens and has been used as a biomarker for diagnosis of invasive aspergillosis. BALF galactomannan test has a sensitivity and specificity of about 38–50% and 87–100% for the diagnosis of CPA [54,66]. The use of BALF galactomannan tests is challenging in resource-limited settings given that bronchoscopy is not routinely performed. High-volume culture methods, however, have good sensitivity in recovering *Aspergillus* spp. from samples, increasing the chance of identification [67,68]. Multiple sputum samples probably increases culture yield.

### 5.5. Other Relevant Tests

Polymerase chain reaction (PCR) and pyrosequencing are the other methods that can detect polymorphisms of CPA-causing *Aspergillus* spp. and antifungal resistance. These are culture-independent molecular methods with higher sensitivity because they can detect small amounts of DNA of causative agents in the patient samples; regarding detection of CPA-causing *Aspergillus* spp., PCR has an average sensitivity and specificity of approximately 80% [69,70,71].

## 6. Management Approach

Considering CPA represents a spectrum of diseases, its optimal management will vary depending on the clinical presentation of the patient and the radiological phenotype of the disease, i.e., *Aspergillus* nodule, simple aspergilloma, CCPA and CFPA (see below). A multidisciplinary approach is key in the management of patients with CPA. A comprehensive management approach requires a combination of close monitoring, and medical (antifungal therapy, usually oral), surgical and radiological interventions.

The overall management goals in CPA are as follows:To improve symptoms and patients’ ‘functional status’—quality of lifeTo prevent the progressive destruction of lung tissue and the development of pulmonary fibrosisTo arrest or prevent haemoptysisTo prevent the emergence of antifungal resistanceTo avoid antifungal toxicityTo reduce death rates and morbidity

### 6.1. Simple Aspergilloma

In patients with simple aspergilloma, those that are asymptomatic and clinically stable for a period of 6 months to 2 years are continually closely monitored and do not need antifungal treatment; for those that are symptomatic, especially presenting with potentially life-threatening haemoptysis, surgical resection is recommended as a definitive cure, as long as the patients have adequate pulmonary function [42,72]. Aspergillomas can spontaneously resolve, but that has been noted in less than 10% of cases [73]. In patients with poor pulmonary function, surgery on the aspergilloma is mostly contraindicated because of post-operative pulmonary deterioration or broncho-pleural fistulae, which can be fatal [74]. Bronchial artery embolisation (BAE) is therefore employed in these cases as it is a less invasive technique than surgery that can be used as a short-term control against haemoptysis until the patients are fit to undergo surgery, or are started on antifungal therapy [75,76].

Antifungal therapy has limited benefit in the treatment of a simple aspergilloma: it is reserved for symptomatic patients who are unable to undergo surgery, as well as those who are immunocompromised or have radiologic progression [77,78,79,80]. The antifungal regimen used is the same as that of CCPA and CFPA (see below). Antifungal therapy, such as voriconazole, can be given at pre- and post-operative periods to reduce the likelihood of post-operative pleural aspergillosis in case of spillage from the cavity being accidentally opened during surgery [77], and relapse.

Radiological monitoring of patients with aspergilloma is best done with low-dose CT scanning at an estimated interval of every 6–12 months in the immunocompromised and less often in the immunocompetent: this is done to assess disease progression and treatment effectiveness. However, if new symptoms emerge, especially haemoptysis, it is also an indicator for additional imaging and investigation for other infectious agents [77]. A noted persistent elevation of *Aspergillus* IgG titres may suggest treatment failure.

### 6.2. CCPA and CFPA

Patients with CCPA and CFPA that are asymptomatic with stable disease do not require antifungal therapy, but should be carefully monitored with objective parameters (imaging, lung function and *Aspergillus* IgG titers. However, for those who are symptomatic with radiographically and serologically progressive disease, antifungal therapy is indicated [77]. Oral triazoles are the mainstay or therapy, as they are known to be active against *Aspergillus* spp. and are relatively well tolerated; the one exception being fluconazole, which has no effect on *Aspergillus* spp. These drugs act by interfering with cell membrane synthesis in susceptible *Aspergillus* spp. by inhibiting the enzyme lanosterol 14 alpha-demethylase, hence preventing ergosterol conversion from lanosterol [81].

For CCPA therapy, the first-line is oral itraconazole (200 mg twice daily) or voriconazole (150 mg to 200 mg twice daily), both with therapeutic drug level monitoring, which is impractical in most centres in sub-Saharan Africa. The aim of the therapy is to improve or prevent symptoms as well as reduce progression of fibrosis [43,82,83]. At present, the choice between the two is based on cost, availability, toxicity and tolerability; they are therefore used interchangeably as first-line agents [81,82]. Treatment for at least 6 months is sufficient for patients with limited disease; for those with bilateral or extensive disease, 24 months or life-long treatment may be required to avoid relapse following discontinuation of therapy [84]. However, voriconazole is preferred for large aspergillomas and patients with more severe disease (bilateral disease) [85]. Itraconazole has been found to show a mean overall 6-month response rate in CPA cases of 63–76.5%; regrettably, it has been associated with adverse effects in about 40–50% cases: some of these adverse effects include gastrointestinal upset, ankle edema, hypertension, peripheral neuropathy, hair loss, etc. [77,81,86,87]. However, these side effects may be lower in LMICs since the patients are much younger and are more likely to tolerate these drugs [25,87]. Voriconazole has been reported to have a response rate of 64% at 3 months and 32% at 6 months, but has also been noted to have more common occurrences of adverse reactions that include transient and reversible visual disturbance, papilledema, optic neuritis, hepatotoxicity, fluorosis, QTc prolongation and photosensitivity with an associated risk of developing skin malignancies in pale skinned, immunosuppressed people [68,88,89,90].

All triazoles inhibit CYP3A4, and voriconazole also CYP2C9 and CYPC19 and thus have great potential for drug–drug interactions with several over-the-counter drugs that can lead to medical complications in patients [91,92,93].

Intravenous drug therapy singly or in combination with azoles is a last resort following clinical failure, pan-azole resistance or drug toxicity, but can also be used in some situations of severe illness presentation. The intravenous drugs used include amphotericin B or an echinocandin, usually micafungin or caspofungin [43,81]. However, due to the significance of the nephrotoxicity associated with the intermittent or long-term use of amphotericin B, some practitioners prefer to use echinocandins [94,95]. One of the authors (DWD) noted that a week course of intravenous amphotericin B does last some patients months and can be followed by itraconazole maintenance therapy, with good disease control. This may be due to the immunological stimulus that amphotericin gives. Micafungin is the echinocandin of choice for CPA therapy because it is the only one that has been extensively studied: its efficacy over 4 weeks is similar to that of voriconazole, reaching up to 68% with reduced side effects and drug interactions; it is therefore safer in the treatment of CCPA and SAIA [94,96,97,98]. Courses given are usually 150–300 mg/day for a duration of 3–12 weeks [94,97,98]. Caspofungin has similar efficacy to micafungin [98].

Surgery, commonly a lobectomy, can be considered in CCPA with severe haemoptysis or failed therapy, but it is associated with a high rate of post-operative complications and the possibility of recurrence [99,100,101]. Patients need to be given antifungals before and peri-operatively, if possible, to avoid post-operative complications.

CFPA requires antifungal therapy, as mentioned for CCPA, and additional supportive treatment of antibiotics for concurrent bacterial infections, oxygen support, physiotherapy, adjunctive interferon-gamma injections, etc. [102,103]. Weekly interferon-gamma injections have been shown to reduce acute exacerbations and hospitalizations for CPA [104].

### 6.3. Aspergillus Nodule

Surgery is the first-line therapy, especially if lung cancer cannot be ruled out, and these patients should not receive antifungal therapy unless symptomatic, but ought to be followed up with serial imaging and *Aspergillus* antibody titres to detect infection relapse [77,101]. For symptomatic patients without access to surgical interventions, treatment is the same as for CCPA/CFPA above.

## 7. Prognosis of Treated CPA

There is no clear consensus on what constitutes treatment success or cure of CPA, yet this would be a fundamental point for estimating survival. Studies have used composite outcomes that include improvement in signs and symptoms, resolution of radiological findings and mycological evaluation [26,87,105]. Chest CT imaging seems to be the only objective measure of treatment response that correlates with clinical response [106]. However, weight gain, improvement of respiratory symptoms and a reduction in the IgG have also been used as treatment endpoints [85].

## 8. CPA Recurrence

After discontinuation of antifungal therapy, relapse occurs in up to 36% of CPA patients with a median duration of 120–160 days and is predicted by bilateral pulmonary disease and involvement of more than one lobe [84,107].

Relapse/recurrence of CPA has been associated with younger age, multi-lobar disease, slow radiological resolution, and prolonged therapy with antifungal drugs [87]. In a study done by Koyama and colleagues, recurrence of CPA was found in about a third of cases post azole treatment discontinuation: the study looked at patients who achieved resolution of clinical, serological and radiographic manifestations of CPA and stopped taking antifungal triazoles [107]. In other studies, it was noted that relapse of CCPA is common after discontinuation of therapy, as is the case with CFPA [87,108]. Aspergillomas have a low recurrence rate. [109].

In studies where patients underwent surgical management of CPA, with or without accompanying anti-fungal therapy and other procedures, recurrence of CPA was found to still be an issue, noted especially with CCPA. [101]. There is indeed need for more large-scale detailed research into recurrence of CPA to get a clearer picture of it.

## 9. CPA Mortality and Predictors

There are limited data on studies from resource-limited settings that have reported survival data among patients with CPA. In the UK, the 1-, 5- and 10-year survival in intensively treated patients has been reported to be 86%, 62% and 47%, respectively [39]. In Japan, a similar estimate of 83% after 1 year and 61% after 5 years is reported among patients who had a history of pulmonary NTM disease [110]. However, Ohba et al. reported a 50% mortality among patients with CPA in a small study in Japan [33]. CPA survival is affected by the clinical and radiological phenotypes of the patients. Patients with SAIA are reported to have a poor prognosis, with 51% of patients dying within 18 months [111]. Conversely, the 2-year survival among patients with CCPA was 72.4% and was comparable to survival among patients with simple aspergilloma [112]. However, following surgical intervention, the 10-year survival of simple pulmonary nodule is 69–90% while that of CCPA is approximately 63–80% [101]

Studies reporting predictors of mortality among patients with CPA are limited by small sample sizes. In Iran, predictors of mortality were diabetes, hypoxic respiratory failure and length of hospital stay [113]. Lowes et al. reported mortality to be associated with NTM, COPD, low albumin, higher age, a low activity score on the St. George Respiratory Questionnaire (SGRQ) and imaging findings consistent with bilateral aspergillomas, cavitary disease and pleural involvement in a large study in the UK [39]. Among patients with NTM-related CPA, an elevated C-reactive protein and the use of corticosteroids seem to predict mortality [110]. Other predictors of mortality reported, albeit in small studies, are low body mass index, a higher total leucocyte count, a lower platelet count and elevated liver enzymes [114].

## 10. Conclusions

In conclusion, CPA occurs in patients with underlying lung diseases such as previous TB, atypical mycobacteria, COPD, sarcoidosis or lung cancer. Its clinical presentation and radiological features overlap substantially with the underlying diseases themselves. Long-term oral triazole therapy is the mainstay for the management of CPA, with itraconazole and voriconazole both being effective. Key challenges with long-term triazole therapy include, antifungal resistance, adverse events, intolerance and drug–drug interactions. Intravenous amphotericin B or echinocandins are useful after failure of oral azole therapy. Relapse of CPA symptoms are common following discontinuation of long-term maintenance therapy.

## Figures and Tables

**Figure 1 jof-06-00075-f001:**
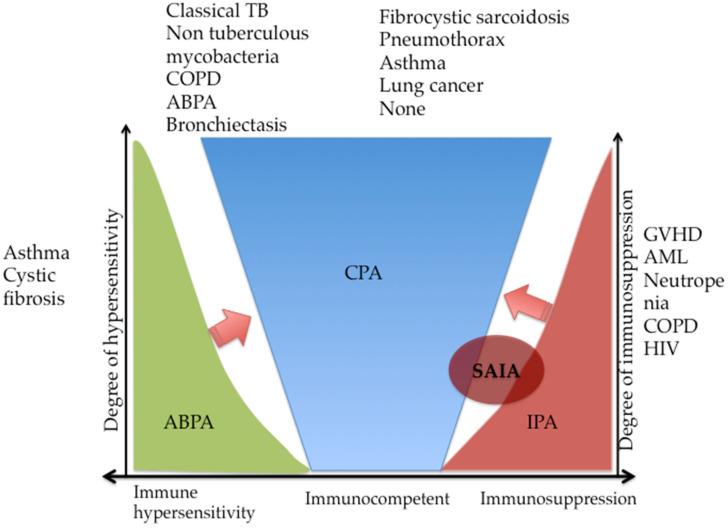
Spectrum of pulmonary aspergillosis based on the host-pathogen damage framework. ABPA, allergic bronchopulmonary aspergillosis; CPA, chronic pulmonary aspergillosis; IPA, invasive pulmonary aspergillosis; SAIA, subacute invasive pulmonary aspergillosis; GVHD, graft-versus-host disease; AML, acute myeloid leukaemia; COPD, chronic obstructive pulmonary disease; HIV, human immunodeficiency virus; TB, tuberculosis. ABPA and IPA can progress to CPA (arrows).

**Figure 2 jof-06-00075-f002:**
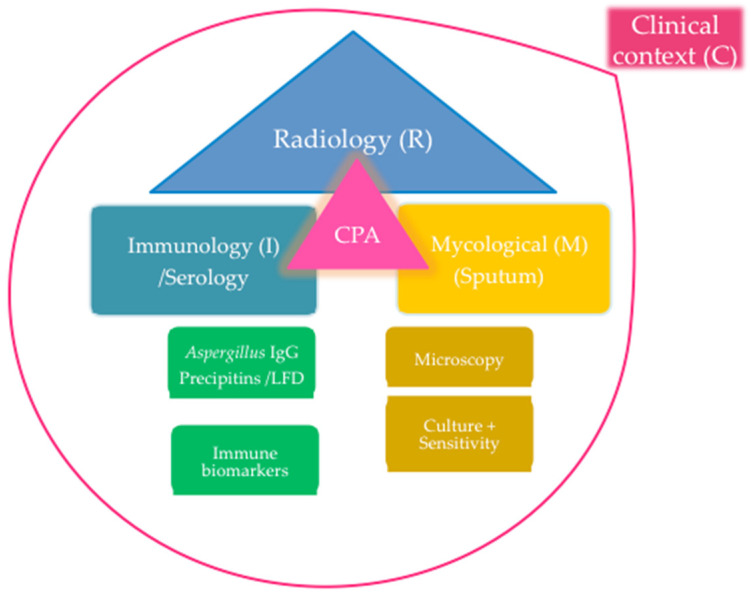
The Clinical, Radiological, Immunological and Mycological (CRIM) approach to the diagnosis of chronic pulmonary aspergillosis. LFD, lateral flow device; CPA, chronic pulmonary aspergillosis; PCR, polymerase chain reaction. The red circle emphasizes the fact that clinical, radiological and microbiological findings should be interpreted with respect to the patient’s clinical presentation.

**Figure 3 jof-06-00075-f003:**
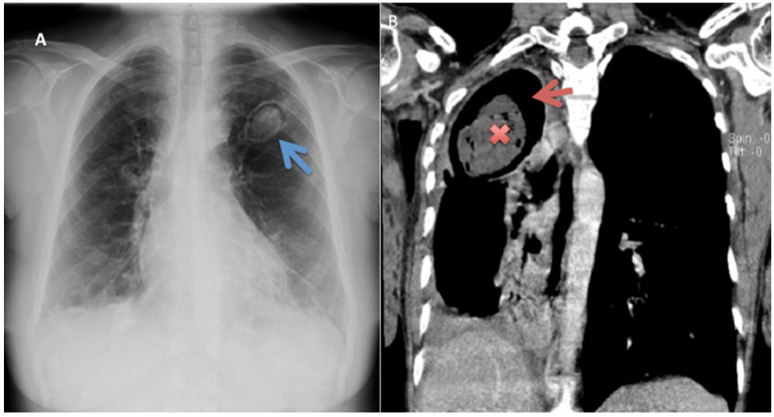
A chest X-ray of a 56-year-old woman with previously treated pulmonary tuberculosis (PTB) showing left apical thick-walled cavity with a well-circumscribed fungal ball (blue arrow) (**A**). A chest computed tomography of a 45-year old man with underlying diabetes and previously treated for PTB showing a fungal ball (×) and an air-crescent sign (red arrow), typical of chronic cavitary pulmonary aspergillosis (**B**). Images courtesy of Dr. Felix Bongomin.

**Figure 4 jof-06-00075-f004:**
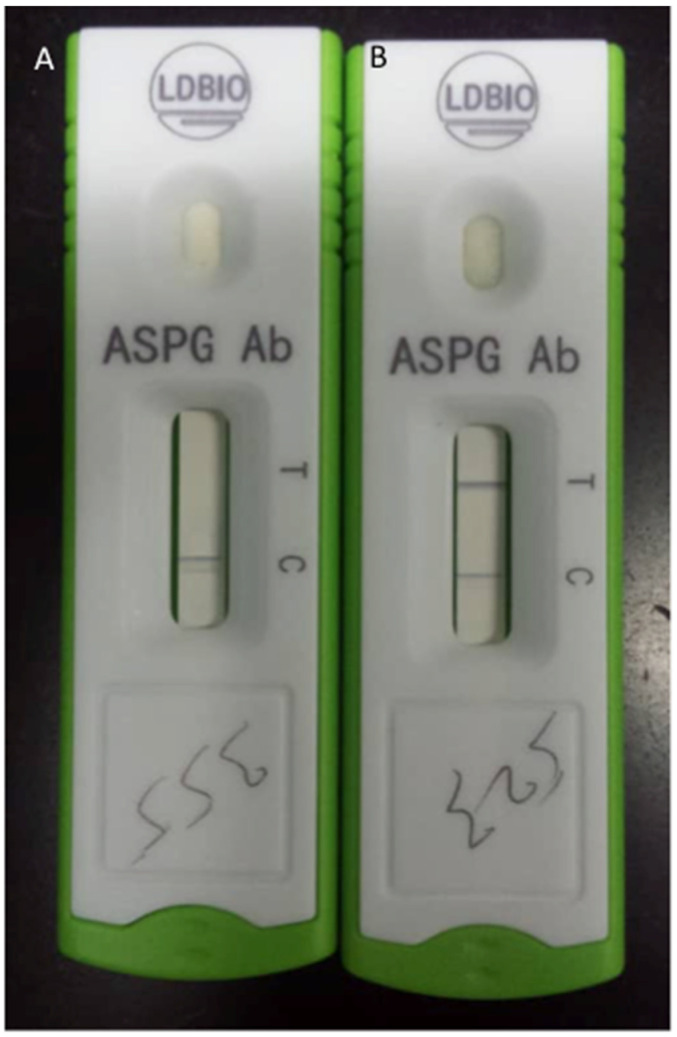
A negative (**A**, single line) and a positive (**B**, double line) *Aspergillus*-specific IgM/IgG in a patient with a suspected chronic pulmonary aspergillosis. Images courtesy of Mr. Richard Kwizera.

**Table 1 jof-06-00075-t001:** Findings of epidemiological studies on chronic pulmonary aspergillosis.

Author (Reference)	Year	Country	Population	CPA Prevalence	Comment
Page et al. [30]	2019	Uganda	284 patients previously treated for PTB	6.3%	CPA was significantly more common in those with chest radiography cavitation (26% vs. 0.8%) and less frequent in HIV co-infected patients (3% vs. 6.7%)
Oladele et al. [25]	2017	Nigeria	208 patients at end of TB treatment or being treated for smear-negative PTB	8.7%: 6.5% in HIV-positive and 14.5% in HIV-negative	153 (73.6%) were HIV-positive
Hedayati et al. [31]	2015	Iran	124 patients with TB (94 current and 30 previous TB)	13.7%: 2.4% aspergilloma and 14% CCPA	38.7% had residual cavities after TB

COPD, chronic obstructive pulmonary diseases; CCPA, chronic cavitary pulmonary aspergillosis.

**Table 2 jof-06-00075-t002:** Underlying conditions in chronic pulmonary aspergillosis.

Underlying Condition	Frequency
Tuberculosis	17–80%
COPD ± Emphysema	30–50%
NTM infection	<20%
Pneumothorax or bullous lung disease	9–20%
ABPA	12–18%
Pulmonary fibrocystic sarcoidosis	9–17%
Lung irradiation	~5%
Rheumatoid arthritis	2–4%
Ankylosing spondylitis	<5%
None	2–10%

ABPA, allergic bronchopulmonary aspergillosis; COPD, chronic obstructive pulmonary aspergillosis. Data from Denning et al. [40] and Smith et al. [27].

**Table 3 jof-06-00075-t003:** Diagnostic criteria for chronic pulmonary aspergillosis.

Criteria	ESCMID/ERS/ECMM [42]		IDSA [43]	GFIF II (GAFFI) [32]	
1	One or more cavities with or without a fungal ball present or nodules on thoracic imaging	All present for ≥3 months	Three months of chronic pulmonary symptoms or chronic illness or progressive radiographic abnormalities, with cavitation, pleural thickening, pericavitary infiltrates and sometimes a fungus ball	Weight loss, persistent cough and/or haemoptysis	All present for ≥3 months
2	Direct evidence of *Aspergillus* infection or an immunological response to *Aspergillus* spp.	-	*Aspergillus* IgG antibody elevated or other microbiological data	A positive *Aspergillus* IgG assay result or other evidence of *Aspergillus* infection.	*-*
3	Exclusion of alternative diagnosis	-	No or minimal immunocompromise, usually with one or more underlying pulmonary disorders	Chest images showing progressive cavitary infiltrates and/or a fungal ball and/or pericavitary fibrosis or infiltrates or pleural thickening; and	-

CPA: Chronic pulmonary aspergillosis; ESCMID/ERS/ECMM: European Society for Clinical Microbiology and Infectious Diseases, the European Respiratory Society and the European Confederation of Medical Mycology; IDSA: Infectious Diseases Society of America; GFIF, Global Fungal Infection Forum; GAFFI, Global Action Fund for Fungal Infections.

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
