# Peer review of "Chronic Pulmonary Aspergillosis: Notes for a Clinician in a Resource-Limited Setting Where There Is No Mycologist"

_jof, 2020, doi:10.3390/jof6020075_

Round 1

Reviewer 1 Report

The manuscript reviews the diagnosis and management of Chronic Pulmonary aspergillosis and attempts to apply these to resource limited settings. It is generally well written easy to follow.

I think the manuscript would benefit from including a minimum requirement testing/management strategy, for application to resource limited settings. Figure 2 is included but is quite generic and includes things like PCR, which probably have a restricted role when resources are limited. While the manuscript also lists many tests and strategies it needs to relate these more to resource limited settings.

I have the following specific comments for the attention of the authors:

  1. Abstract: The preferred use of voriconazole over itraconazole due to heart failure seems a little counter intuitive given the association between voriconazole and prolongation of the QT interval.
  2. Abstract Lines 28-29: As written it states that you are treating TB with azoles! Please revise the sentence
  3. Introduction, Line 46: Please add that these syndromes can be quite fluid if the individuals immune status changes and also state that while the immune status dictates what form of Aspergillosis the patient is at risk from, it also dictates whether or not they are likely to get Aspergillosis at all.
  4. Definition, Line 89; Please insert “inevitable prior to “repeated exposure”
  5. Epidemiology Line 108: States Table 2, should be table 1? Also Please provide some data for countries where the incidence of TB is less, to support your statement in line 102.
  6. Underlying conditions Line 121: The underlying conditions do not “cause” CPA, the causative agent is Aspergillus, these are clinical risk factors. Please amend.
  7. Line 133: Please spell out C.R.I.M in the heading
  8. Table 3: Due to a formatting issue with the PDF I am unable to read all of table 3.
  9. Figure 2: Please confirm this is directed as resource limited settings, if so amend the heading to reflect this.
  10. Radiology: Move last paragraph (lines 44-47) to the first paragraph, as it describes the preferred the radiological investigation. Line 42: lung cancer is one amongst several differential diagnoses indicated by a nodule.
  11. As with the section on management of CPA the manuscript needs a section describing the clinical presentation of each of the CPA manifestations (Aspergilloma, CCPA, CFPA, nodule, SAIA). Maybe the radiological section could be amended to provide this?
  12. Immunology Line 74: What is the preferred assay, immunocap?
    Line 81: Insert specifically between test and for A. fumgiatus. How accessible are these non-fumigatus antibody tests?
  13. Microbiology, Lines 100-101: Given the poor sensitivity of microscopy I would never use to determine whether other procedures were performed. Culture should be performed irrespective of the result. Also please confirm that you are talking about the testing of respiratory samples.
  14. Microbiology Line 106: Galactomannan has not been mentioned previously. Please provide some background. Also is there any role for LFA detecting antigen in BAL samples in resource limited settings?
  15. Treatment Lines 176: prolonged use of voriconazole may also increase the likelihood of side-affects, particularly important for the management of chronic conditions. The prolongation of the QT interval also needs to be listed.
  16. Aspergillus nodule: Is antifungal treatment the same as CCPA/CFPA?
  17. CPA Recurrence, line 221, CNPA, you have not mentioned this previously, it is assumed that this should be SAIA?
  18. Mortality/Predictors, Lines 231-232. Sentence “CPA…………….syndrome” is unclear, please revise.

Author Response

Many thanks for these very useful comments.

We have addressed all the comments.

A revised figure 2 has been uploaded.

Responses to Reviewer#1

The manuscript reviews the diagnosis and management of Chronic Pulmonary aspergillosis and attempts to apply these to resource limited settings. It is generally well written easy to follow.

I think the manuscript would benefit from including a minimum requirement testing/management strategy, for application to resource limited settings. Figure 2 is included but is quite generic and includes things like PCR, which probably have a restricted role when resources are limited. While the manuscript also lists many tests and strategies it needs to relate these more to resource limited settings.

I have the following specific comments for the attention of the authors:

  1. Abstract: The preferred use of voriconazole over itraconazole due to heart failure seems a little counter intuitive given the association between voriconazole and prolongation of the QT interval.

Authors’ response: Thank you, this has been revised

  1. Abstract Lines 28-29: As written it states that you are treating TB with azoles! Please revise the sentence

Authors’ response: Thank you, this has been revised

  1. Introduction, Line 46: Please add that these syndromes can be quite fluid if the individuals immune status changes and also state that while the immune status dictates what form of Aspergillosis the patient is at risk from, it also dictates whether or not they are likely to get Aspergillosis at all.

Authors’ response: Thank you, we have added a sentence

  1. Definition, Line 89; Please insert “inevitable prior to “repeated exposure”

Authors’ response: Thank you, this has been inserted

  1. Epidemiology Line 108: States Table 2, should be table 1? Also Please provide some data for countries where the incidence of TB is less, to support your statement in line 102.

Authors’ response: Thank you, in Table 2; we only included original studies involving a population of TB patients only. Such studies have not been done in low TB incidence countries, besides the country estimates

  1. Underlying conditions Line 121: The underlying conditions do not “cause” CPA, the causative agent is Aspergillus, these are clinical risk factors. Please amend.

Authors’ response: Thank you, this has been revised

  1. Line 133: Please spell out C.R.I.M in the heading

Authors’ response: Thank you, this has been revised

  1. Table 3: Due to a formatting issue with the PDF I am unable to read all of table 3.

Authors’ response: Thank you, this is a formatting error the table is complete.

  1. Figure 2: Please confirm this is directed as resource limited settings, if so amend the heading to reflect this.

Authors’ response: Thank you,  we agree , we deleted some HIC tests

  1. Radiology: Move last paragraph (lines 44-47) to the first paragraph, as it describes the preferred the radiological investigation. Line 42: lung cancer is one amongst several differential diagnoses indicated by a nodule.

Authors’ response: Thank you, this has been revised

  1. As with the section on management of CPA the manuscript needs a section describing the clinical presentation of each of the CPA manifestations (Aspergilloma, CCPA, CFPA, nodule, SAIA). Maybe the radiological section could be amended to provide this?

Authors’ response: Thank you, the radiology section has a brief description of these CPA phenotypes

  1. Immunology Line 74: What is the preferred assay, immunocap?

Authors’ response: Thank you, Yes. But we advocate for LFD in Low resource setting

Line 81: Insert specifically between test and for A. fumgiatus. How accessible are these non-fumigatus antibody tests?

Authors’ response: Thank you, this is not very accessible in LMICs

  1. Microbiology, Lines 100-101: Given the poor sensitivity of microscopy I would never use to determine whether other procedures were performed. Culture should be performed irrespective of the result. Also please confirm that you are talking about the testing of respiratory samples.

Authors’ response: Thank you, this has been revised

  1. Microbiology Line 106: Galactomannan has not been mentioned previously. Please provide some background. Also is there any role for LFA detecting antigen in BAL samples in resource limited settings?

Authors’ response: Thank you, this has been revised. BAL LFA has very low diagnostic performance. Moreover, obtaining a BAL sample is not routine in LICs

  1. Treatment Lines 176: prolonged use of voriconazole may also increase the likelihood of side-affects, particularly important for the management of chronic conditions. The prolongation of the QT interval also needs to be listed.

Authors’ response: Thank you, this has been revised

  1. Aspergillus nodule: Is antifungal treatment the same as CCPA/CFPA?

Authors’ response: Thank you, this has been revised. On for symptomatic patients who do not have access to surgical interventions

  1. CPA Recurrence, line 221, CNPA, you have not mentioned this previously, it is assumed that this should be SAIA?

Authors’ response: Thank you, this has been revised

  1. Mortality/Predictors, Lines 231-232. Sentence “CPA…………….syndrome” is unclear, please revise.

Authors’ response: Thank you, this has been revised

Best regards

Dr. Bongomin

Reviewer 2 Report

Well written and comprehensive review.

1.Line 28 :please add  " patients Co- infected" with TB to make the point about rifampin-Azole interaction more clear

2.Table in page 6 is not completely visible

3.In discussing risk factor, special mention of CGD associated aspergillosis is worth mention

4.Line 173 : may also add fluoride toxicity to voriconazole toxicities

5.Lines 181-199: can you comment on the role of combination therapy

6.Can you comment on the use of interferon gamma replacement as an adjunctive therapy consider reviewing this reference .PMID: 32229542

Author Response

Many thanks for these very useful comments.

We have addressed all the comments.

We added a statement on IFN gamma therapy

Responses to Reviewer#2:

Well written and comprehensive review.

  1. Line 28: please add  " patients Co- infected" with TB to make the point about rifampin-Azole interaction more clear

Authors’ response: Thank you, this has been revised

  1. Table in page 6 is not completely visible

Authors’ response: Thank you, this is due to formatting error. Otherwise the table is complete. With published 3 diagnostic criteria; IDSA, ERS and GAFFI

  1. In discussing risk factor, special mention of CGD associated aspergillosis is worth mention

Authors’ response: Thank you, we have included CGD

  1. Line 173: may also add fluoride toxicity to voriconazole toxicities

Authors’ response: Thank you, We have added fluorosis.

  1. Lines 181-199: can you comment on the role of combination therapy

Authors’ response: Thank you, this has been revised; Though not routine, IV and azoles have been used in combinations.

  1. Can you comment on the use of interferon gamma replacement as an adjunctive therapy consider reviewing this reference .PMID: 32229542

Authors’ response: Thank you, we have added the findings from this very interesting study.

Best regards

Dr. Bongomin
